# Hepatocellular Carcinoma—The Influence of Immunoanatomy and the Role of Immunotherapy

**DOI:** 10.3390/ijms21186757

**Published:** 2020-09-15

**Authors:** Keyur Patel, Ryan Lamm, Peter Altshuler, Hien Dang, Ashesh P. Shah

**Affiliations:** 1Department of Surgery, Thomas Jefferson University, Philadelphia, PA 19144, USA; Keyur.Patel@jefferson.edu (K.P.); Ryan.Lamm@jefferson.edu (R.L.); Peter.Altshuler@jefferson.edu (P.A.); 2Sidney Kimmel Cancer Center, Philadelphia, PA 19107, USA

**Keywords:** hepatocellular carcinoma, tumor microenvironment, immunoanatomy, immunotherapy, CAR T cell

## Abstract

Hepatocellular carcinoma (HCC) is a leading cause of cancer-related morbidity and mortality worldwide. Most patients are diagnosed with advanced disease, limiting their options for treatment. While current treatments are adequate for lower staged disease, available systemic treatments are limited, with marginal benefit at best. Chimeric antigen receptor (CAR) T cell therapy, effective in treating liquid tumors such as B-cell lymphoma, presents a potentially promising treatment option for advanced HCC. However, new challenges specific to solid tumors, such as tumor immunoanatomy or the immune cell presence and position anatomically and the tumor microenvironment, need to be defined and overcome. Immunotherapy currently in use must be re-engineered and re-envisioned to treat HCC with the hopes of ushering in an answer to advanced stage solid tumor disease processes. Future therapy options must address the uniqueness of the tumors under the umbrella of HCC. This review strives to summarize HCC, its staging system, current therapy and immunotherapy medications currently being utilized or studied in the treatment of HCC with the hopes of highlighting what is being done and suggesting what needs to be done in the future to champion this therapy as an effective option.

## 1. Introduction

Hepatocellular carcinoma (HCC) is the fourth most common cause of cancer-related mortality and the leading cause of cancer-related morbidity worldwide [1,2,3]. Relative to other malignancies, the incidence of HCC is fifth highest in men and ninth highest in women [4]. Risk factors for these patients are numerous including—hepatitis B virus (HBV), hepatitis C virus (HCV), non-alcoholic steatohepatitis (NASH), alcoholic liver disease, tobacco, aflatoxins, vinyl chloride and thorium dioxide, anabolic steroids, as well as other rare diseases affecting the liver and environmental toxins [5,6]. Advances in preventative measures against conditions leading to HCC, such as the HBV vaccine, have reduced its incidence, while antiviral therapy in HCV has been able to achieve sustained virologic responses and disease suppression. Despite these advances, in 2018, global mortality related to HCC still exceeded 600,000 deaths and is expected to rise to over 1 million by 2030 [7,8].

Unfortunately, early diagnosis is rare due to the vague symptoms associated with liver tumors and lack of effective identification and screening of non-cirrhotic at-risk populations. These shortcomings are highlighted by the geographic discrepancy in expert opinion; currently, the United States and Canada do not recommend any screening owing to a lack of conclusive evidence proving a mortality benefit in these populations [9,10]. Conversely, the European Association for the Study of the Liver (EASL) guidelines support routine 6-month ultrasound surveillance in non-cirrhotic HBV populations or those with advanced fibrosis, supported by a 2011 meta-analysis of 15,158 patients showing an association between screening and improved overall survival through early disease detection [11,12]. Similarly in Asia, the Asia Pacific Association for the Study of the Liver (APASL) recommend both serologic and radiologic surveillance of patients as young as 20 with HBV surface antigen positivity or those with family history of HCC [13]. Regardless of these heterogeneous screening practices, early detection and treatment with curative intent is uncommon, reflected in the dismal 18% 5-year survival and median survival of 6–20 months from time of diagnosis [3,14]. Moreover, HCC tends to present in its advanced stage necessitating the need for better systemic therapy.

Carcinogenesis in HCC is the result of a variety of ongoing and disparate insults stemming from heterogeneous disease processes. Regardless of the initial insult, HCC arises in a setting of sustained inflammation, generation and inadequate clearance of toxic reactive oxygen species and a host of genomic alterations all causing hepatocellular stress and dysfunction. These ongoing processes disrupt normal hepatocellular function and lead to several overarching processes, including progressive genomic instability, cell death and fibrosis of normal hepatocytes leading to cirrhosis and abnormal hepatocellular regeneration causing dysplastic and neoplastic cellular proliferation each specific to the causative etiology.

While tumorigenesis in the liver stems from an insult that disrupts normal hepatic and hepatocellular structure and function, the liver itself is unique to most other solid organs due to the presence of systematically fixed phagocytotic cells which control bloodstream screening and clearance. This is known as the reticuloendothelial system (RES) or, more recently, the mononuclear phagocyte system (MPS) [15]. These terms are, somewhat incorrectly, used interchangeably but the latter, more modern term, refers to a phagocytic system which is fixed and abundant mainly in the liver, spleen and lymph nodes [15]. This unique immune-environment in the liver, which we refer to as the immunoanatomy, plays a vital role in creating conditions utilized by HCC tumor cells.

Most current standard treatments for advanced stage HCC are associated with significant systemic side effects and a modest at best survival benefit [16]. Advances in immunotherapy, however, have shown success in treating tumors across multiple organ systems including the lung, kidney, bladder, lymphoid tissue and skin. As such, it may provide a future therapeutic option for HCC patients. In this review, we discuss current diagnosis and treatment of HCC, the unique contribution of the immunoanatomy of the liver in both its natural healthy and HCC disease state and provide an overview of immunotherapy in HCC from the past and present with the hope of shedding light on where the field needs to expand in the future.

## 2. Etiologies Predisposing to HCC

While HCC is defined as a singular entity, the liver pathologies resulting in HCC carcinogenesis are highly variable depending on the inciting injuries. Each insult utilizes a different process to promote genomic instability, ongoing inflammation, cellular damage and neoplastic proliferation. Here, we will provide an overview of each of the main disease processes in the liver that may progress to HCC.

Most cases of HCC are due to viral causes, namely hepatitis B virus (HBV). HBV is a partially double-stranded DNA virus and can either exist as an acute infection with longstanding immunity or a chronic infection with the potential for reactivation [17]. HBV causes HCC by directly damaging hepatocytes leading to deranged regeneration as well as by integrating into the hepatocyte genome and altering transcription, translation and regulation [18]. The direct insult, acute viral infection of hepatocytes, activates the highly immunogenic liver immunoanatomy to recruit immune response cells (i.e., T cells, B cells, macrophages, etc.). The result is inflammation, degeneration and regeneration of the hepatocytes [18]. In chronic infection, on the other hand, integration of the HBV viral genome results in immune tolerance of HBV and HBV related molecules in the liver, possibly via inhibition or decreased expression of a co-inhibitory receptors on T cells that mitigate the CD8 T cell response [19]. Likely this process is carried out via multiple immune- and genome-modulating pathways yet to be discovered. Interestingly, the specific immunoanatomy even within HBV can be augmented and/or modified from other etiologies. For example, aflatoxin exposure has been shown to have a synergistic effect when combined with HBV, leading to a higher incidence of HCC in patients exposed to both [20]. In the setting of inflammatory regeneration leading to fibrosis and immune tolerance, the liver becomes vulnerable to HCC neoplastic proliferation.

Alternatively, HCV, which is an RNA virus, cannot integrate into the host hepatocyte genome and thus requires continued replication [21]. This continuous replication causes chronic inflammation leading to the degeneration and fibrosis as described above. However, evidence of differing tumorigenesis and immunoanatomy based on immune cell types present between HBV and HCV continues to evolve as our understanding of HCC increases [21]. Overall, the chronic, immune-driven inflammation characteristic of these viral etiologies leads to or creates the milieu that results in the majority of HCC cases.

Another etiology, non-alcoholic steatohepatitis (NASH), is a growing epidemiologic burden with the rising prevalence of obesity in the US and worldwide. NASH causes inflammation in hepatocytes by depositing excess circulating fat globules within the cells. These fatty hepatocytes become distressed and compete for oxygen leading to inflammation, loss of function and an immune response that results in subsequent fibrosis [22]. NASH also causes “multiple-parallel” hits whereby fatty deposits damage and inflame visceral tissues which cause increased metabolic stress in the already weakened fatty hepatocytes tasked with filtering the molecules of inflammation combined with a globally poor nutritional state [23]. Specifically, Kupffer cells, which are specialized macrophages located within the liver sinusoids and part of the normal healthy liver immunoanatomy, are activated, cytokine production is increased and an environment of fibrosis, regeneration and deranged proliferation is created [24].

Alcohol is a well-known carcinogen for a variety of cancers and contributes to their development in a multimodal fashion. Acetaldehyde, the metabolite of ethanol, impairs the cell’s DNA repair mechanisms through direct cross-linking [25]. Alcohol also leads to the production of reactive oxygen species, pro-inflammatory cytokines and the downregulation of normal immunosuppression during metabolism, a process that takes place within hepatocytes [26,27,28]. These processes combine to damage and alter hepatocytes, predisposing these cells to mutagenesis and subsequent carcinogenesis.

While each etiology shares the overall theme of insult leading to fibrosis, degeneration, regeneration and finally mutagenesis ending in HCC tumor cells, the process by which these stages are carried out is clearly quite heterogeneous. As such, it becomes extremely important to strive to tailor treatment options based on the specific causative etiology, a task which, as we will see, has yet to be achieved.

## 3. Current Management of HCC

### 3.1. BCLC Staging and Treatment Algorithm

Current HCC treatment pathways depend on the stage at which a patient is diagnosed. The most commonly used staging system in HCC is the Barcelona clinic liver cancer (BCLC) system. The BCLC system classifies patients into five prognostic groups in order of advancing disease stage—0, A, B, C and D. The staging system incorporates liver function (Child-Pugh class), tumor burden (number, size, vascular invasion, metastases) and patient performance status (using the Eastern Cooperative Oncology Group [ECOG] status) [29]. Patients classified as BCLC stage 0 or A are eligible for curative therapies, namely surgical resection, liver transplantation and percutaneous radiofrequency ablation. Five-year survival rates are quoted at greater than 70%. BCLC stage B patients are eligible for locoregional therapy, such as trans-arterial chemoembolization (TACE) or trans-arterial radioembolization (TARE). Median survival is quoted at 2 years. BCLC stage C patients have advanced disease defined as either having portal vein invasion or extrahepatic spread. These patients can be offered systemic treatment only. BCLC stage D, also called terminal stage, is currently treated symptomatically with best supportive care [30]. Figure 1 displays the BCLC staging system with a graphic representation of the associated anatomy, tumor burden liver and recommended treatment. Noticeably, none of the available treatment options addresses the etiology of HCC tumor development. In addition, currently no immunotherapy is considered standard of care for any stage.

### 3.2. Surgical Management

Surgical management of HCC is divided into two categories, resection or surgical removal of part of the liver based on anatomic segments or transplant, the complete removal of the native liver and replacement from deceased- or live-donor tissue. The decision to resect depends on tumor size, location and degree of underlying hepatic decompensation. Patients eligible for resection often have preserved liver function and tumors which are localized anatomically without any vascular invasion, falling into BCLC categories 0 and A [31,32,33]. A resection is potentially curative if tumors are solitary and generally < 5 cm in diameter [31]. Diminished liver function can negate eligibility for resection of HCC tumors, even those that fit the size, location and invasion criteria. Though defined by overriding guidelines, decisions to resect can vary between institutions. Recovery from resection is dependent on the volume of remnant liver and its underlying function [32]. Resection for HCC has a role in the treatment algorithm but does not offer a solution to patients with high tumor burden, advanced HCC, and/or diminished underlying liver function.

Transplant for HCC is usually reserved for patients in BCLC categories 0 and A who do not meet the above resection criteria. In the US, the Milan criteria is often used to determine transplant eligibility for HCC patients. The Milan criteria states that patients who have—(1) a single tumor with diameter less than 5 cm OR (2) not more than three foci of tumor, each one not exceeding 3 cm AND (3) no angioinvasion AND (4) no extrahepatic involvement are eligible for liver transplantation [34,35,36]. In the US, liver transplant is considered the gold standard for treatment of HCC that falls within the Milan criteria [32]. Additional, more liberal criteria proposed by the University of San Francisco expanded the Milan criteria to—(1) a single tumor with diameter less than 6.5 cm OR (2) not more than three foci of tumor, each one not exceeding 4.5 cm OR (3) total tumor diameter not exceeding 8.5 cm AND (4) no angioinvasion [31,37]. Survival at 1, 3 and 5-years were predictably worse in the UCSF group compared to Milan, although only late-stage T4 tumors were shown to have inferior outcomes compared to Milan criteria on multivariable analysis [38]. The use of the Milan criteria, namely in the US, is likely here to stay and is driven partially by ongoing organ shortages and lack of the robust living-donor liver programs which exist in other countries. This limits the number of patients who receive liver transplants. In addition, surgical intervention is invasive and exposes patients to the risks of general anesthesia as well as a gamut of surgical complications highlighting the need for less invasive, more specific treatments.

Radiofrequency ablation (RFA) is being increasingly utilized to treat small, early stage BCLC 0 and A tumors as well as to treat patients who are at risk to progress while waiting on the transplant list [39]. RFA is the superheating of HCC tumor tissue via radio wave transmission through a probe inserted through the skin guided by ultrasound or cross-sectional imaging [39]. A recent study, however, has shown that, in HCC patients who meet the resectability criteria treated with resection versus RFA, the overall 5-year survival and recurrence-free survival were lower in the RFA group (75 v 54% and 51 v 28%, respectively) [40]. This modality, in its current state, is clearly reserved for patients with small tumors who cannot tolerate surgical intervention or those on the transplant waiting list who are at risk of tumor progression but does not offer a solution to patients with advanced HCC, nor is it superior to current surgical options available.

### 3.3. Locoregional Therapy 

As HCC BCLC stage progresses to B, the patient is often no longer eligible for surgical resection, transplant or ablation. Instead, reducing tumor burden becomes the goal of treatment. This scenario is also the case for patients who may be in lower BCLC stages but their diminished overall health renders them ineligible for safe surgical intervention. Treatments designed for this purpose include the direct administration of either intra-arterial chemotherapy (trans-arterial chemoembolization (TACE)) or radiotherapy (trans-arterial radioembolization (TARE)) and are collectively referred to as locoregional therapy [41,42]. Specifically, TACE involves direct delivery of doxorubicin followed by the blocking of blood flow via selective hepatic artery branch embolization to ensure chemotherapy dwell time. Recent advances allow TACE treatment of both early, non-surgical and some late stage patients (BCLC 0, A, B, & some C) with a solitary nodule or up to 3 nodules under 3 cm [41]. One trial, stopped early, showed that mean survival was significantly longer with TACE compared to symptomatic treatment alone (28.6 months v 17.9 months) [42].

TARE, on the other hand, utilizes selective intra-arterial injection of radioactive yttrium-90 (Y-90), iodine-131 (131I) or rhenium-188 (188Re) to cause radiation-induced cell necrosis [43]. Similar to TACE, TARE is indicated for non-surgical BCLC Stage 0, A, B & some C patients [44]. One study reported median survival of 16.9 months in BCLC Stage B patients and another retrospective analysis demonstrated a median survival of 14 months when TARE was used as first-line therapy compared to 8 months in patients receiving standard therapy [44,45]. Most complications of TARE stem from radiation damage causing bile duct stricture and cholangitis.

These locoregional methods of treatment for advanced disease, while showing some benefit to patients, ultimately do not lead to a sustained response. Residual tumor and local recurrence often necessitate additional TARE or TACE procedures and progression of disease sometimes renders these locoregional therapies ineffective with no response seen [42]. Also, these procedures are provided only at selected centers equipped with skilled providers able to perform both the procedure and post-procedural care. This results in limited availability and a significant healthcare cost. While some believe there is the potential for locoregional therapy to “downstage” cancers, making the patient eligible for resection or transplant, this concept has not yet been accepted into standard practice.

### 3.4. Systemic Therapy

For advanced staged HCC tumors (BCLC B & C), surgical and locoregional therapies are not recommended. Instead, the disease is so advanced only systemic therapies are applicable in this patient population. Sorafenib, one of the medications which is FDA approved for advanced HCC, is a systemic oral therapy which inactivates multiple kinase proteins, including vascular endothelial growth factor receptor (VEGFR), platelet derived growth factor receptor (PDGFR) and rapidly accelerated fibrosarcoma (RAF) kinases, halting pathways responsible for angiogenesis and cell growth [46,47]. It is approved for use in the treatment of kidney and thyroid cancer. Sorafenib was tested in late stage HCC (BCLC B & C) treatment in the US & European Sorafenib in patients with advanced hepatocellular carcinoma (SHARP) trial and showed an increased survival versus placebo (10.7 months v 7.9 months) [16]. This finding was corroborated in a repeat trial done in the Asia-Pacific (6.5 months v 4.2 months) [48]. Subsequently, other systemic therapies have been marketed as non-inferior to sorafenib including regorafenib (RESOURCE trial) [49,50], cabozantinib (CELESTIAL trial) [51] and lenvatinib [49]. Although these medications were the first to offer treatment to late stage HCC patients, at best they offered a modest 3 month median survival benefit. The shortcomings of sorafenib, beyond its marginal benefit to the patient, include the frequent intolerance of side effects associated with treatment. Other systemic medications have more recently been approved for HCC, however further studies will be required to elucidate their efficacy. No current standard therapeutic modality offers favorable survival outcomes to patients with advanced stage HCC. However, given the unique nature of the liver and HCC tumors, immunotherapy may be the answer.

## 4. Immune Landscape of HCC

### 4.1. Liver Immunoanatomy

In order to understand how immunotherapy could work in treating HCC, it is first important to characterize the nuanced complexity behind liver immunogenicity in a healthy state. Liver parenchyma is characterized by a rich, dense network of hexagonal lobules consisting of a triad of biliary, arterial and venous structures surrounding a central vein. Within these lobules, 60% of liver parenchyma is composed of hepatocytes. Amongst the remaining tissue are sinusoids lined with Kupffer cells. The liver serves numerous critical functions such as protein synthesis, metabolism and detoxification. The ultimate function of the Kupffer cells is to phagocytose cellular debris from these processes and remove circulating endotoxins, sometimes activating cytokines and recruiting immune cells in response [52]. Between the hepatocytes and sinusoids exists the Space of Disse which house the hepatic stellate cells (HSCs) of Ito. When stressed, HSCs promote fibrosis in response to liver injury or chronic inflammation leading to cirrhosis [53]. These processes are stressed in the setting of liver injury leading to rapid recruitment of immune cells, inflammation, fibrosis, and, ultimately, cirrhosis, setting the stage for mutagenesis and tumorigenesis.

Another critical feature of the liver is its immune regulation and relative immune tolerance [54,55]. Immunomodulation occurs through both innate and adaptive immune responses throughout the organ. The adaptive immune response is initiated by the priming of naïve T cells in the liver. This process involves antigen presentation by antigen-presenting cells (APCs) to a matching T cell receptor and co-stimulatory interaction between B7 ligands on APCs and CD28 receptors on T cells, tailoring responses to antigens in the liver [56,57]. The innate immune response is mainly regulated by the fenestrated liver sinusoidal endothelial cells (LSECs) which express both antigen-presenting molecules, including MHC class I/II and costimulatory CD40, CD80 and CD86 molecules as well as mannose receptors, promoting antigen uptake [58,59]. In this space, proinflammatory cytokines such as IL-2, IFN-γ and TGF-β are constitutively expressed. This cytokine production allows dendritic cells that may otherwise inhibit proliferation to express immunosuppressive ligands such as CTLA-4, PD-1, PDL-1 and PDL-2 during chronic inflammation [60,61]. Expression of anti-inflammatory cytokines downregulates MHC Class II activation and diminishes the innate immune response by reducing T cell activation in the absence of stress. On the other hand, the modulation of T cells during injury can alter the abundant population of natural killer (NK) T cells that facilitate apoptosis and T regulatory cells (Tregs) that maintain cellular homeostasis [62,63,64,65]. It is important to understand these unique features of the liver immunoanatomy to be able to recognize how HCC can hijack them to avoid immune surveillance and promote tumor growth.

### 4.2. Tumor Microenvironment

The HCC tumor utilizes the relative state of immunotolerance afforded to the liver to evade detection, creating what is known as a tumor microenvironment. This microenvironment is comprised of similar immunologic elements as the liver in its healthy state but with differing local recruitment, cellular crosstalk and immunoanatomy. A schematic of normal versus tumor microenvironment is represented in Figure 2. The progressive transformation from healthy liver to tumor microenvironment occurs by continued disturbances to the normal immunologic homeostasis of the liver, notably through secretion of immunosuppressive cytokines, abnormal expression of antigens and changes in the local immune cell crosstalk during states of inflammation [66,67]. Specifically, during periods of acute stress, hepatocytes may alter their own transcriptional profile to release acute phase reactants into the blood or activate apoptotic or necrotic pathways [68,69,70,71]. Regeneration of injured hepatocytes, especially at rapid rates, predisposes the new cells to mutagenesis and tumorigenesis, resulting in new tumor-specific antigens being presented. Activation of these various pathways are etiology-dependent and thus, antigens and receptors are differentially activated or suppressed in response to different etiologies [68,69,70,72].

In addition, there are cancer-specific immune responses. For example, there is the recruitment of tumor-infiltrating lymphocytes (TILs), which normally reside in the tumor cells’ immediate environment or sometimes within the stroma of the tumor [73]. TILs are lymphocytes designed to identify cancer-specific tumor markers. These cells directly or indirectly kill tumor cells by secreting IFN-γ, macrophage activating factors (MAF) and TNF-α or simply by activating apoptotic cascades in tumor cells [34]. These cells contribute to the heterogeneous array of individual tumor microenvironments in HCC. Specific HCC TILs have been identified, however their specificity to individual etiologies and prognostic and therapeutic values have yet to be proven [74,75].

These factors create a variety of unique tumor microenvironments which help explain the heterogeneity of the HCC tumors. In fact, the microenvironments are likely different even amongst HCC tumors resulting from the same etiology. For example, one study showed distinct and differing populations of immune response cells have been found in HBV versus HCV HCC tumor microenvironments [73,76].

Besides the HCC immunomodulation, cancer cells, in general, play a large role in shaping the microenvironment based on their metabolic requirements. Cancer cells require accelerated cellular growth, progressive mutagenesis and altered cellular metabolomics, creating a distinct microenvironment around them to support these requirements characterized by relative hypoxia, acidosis and aerobic glycolysis (known as the Warburg effect) [77]. This creates competition for oxygen and nutrients with neighboring healthy cells, resulting in a hostile and hypoxic environment which leads to further inflammation, fibrosis, immunomodulation and tumor cell creation [78,79,80,81]. As the tumor grows, its microenvironment incorporates more and more nearby non-malignant cells and leads to tumor invasion and metastasis [81,82]. This is the mechanism of HCC progression.

The tumor microenvironment is vital for mutagenesis, tumorigenesis and neoplastic proliferation in HCC. The distinct nature of the microenvironments could provide specific targets, in the form of TILs, modified APCs, activated macrophages or other unidentified antigens for immunotherapeutic regimens [83]. This would unlock what we believe to be the future of etiology- and patient-specific treatments in the form of targeted immunotherapy. While the liver itself does contain some unique features, more research is needed to define how this compares to liver microenvironments of other solid tumors or how it may influence the success or failure of immunotherapy options. Some current treatments have attempted to utilize this mechanism, however many of them come with their own set of limitations.

## 5. Current Immunotherapy Options

### 5.1. Cancer Vaccine

Similar to the development of routine vaccines, vaccines can also be created to prime a patient to target tumor cells. These vaccines can be targeted to specific nucleic acids or polypeptides so that when they present in a patient, the cells will be targeted by the hosts’ immune response [84,85]. There are multiple clinical trials studying the role of vaccines in HCC alone and in combination with various therapies targeting peptides such as alpha-fetoprotein (AFP) (NCT 00005629), a marker normally found in fetal plasma and tested from maternal serum during pregnancy but erroneously secreted by certain HCC tumors with an unknown function [86]. Others include vaccines against VEGFR1 and VEGFR2 (NCT 01266707), transcription factor proteins which promote tumor angiogenesis and against HCC specific stem cells (NCT 02089919). No trials to date have produced any results, so efficacy remains to be seen. However, given the difficulties associated with vaccine creation in general combined with the fact that HCC is both heterogeneous and mutable, we are likely very far from a universally effective HCC vaccine.

### 5.2. Programmed Cell Death Protein 1 (PD-1) Inhibitors

Direct targeting of immunosuppressive receptor-ligand interactions offers a potential method of slowing tumor progression. PD-1 is a T cell receptor which inhibits T cell activation promoting tolerance which, in the tumor microenvironment, allows unchecked tumor proliferation and has been found on TILs in the HCC tumor microenvironment [87]. Previous studies showed that PD-1 expression diminishes HBV viral clearance and induces T cell exhaustion in HCV ultimately allowing for immune cell evasion [88,89,90]. PD-1 inhibitors are small proteins which block interactions between PD-1 and its ligands, preventing the subsequent cell signaling cascade and checkpoint inhibition [87]. PD-1 inhibitors have previously been approved for certain cancers including melanoma and non-small cell lung carcinoma [91]. In a meta-analysis, PD-1 inhibitors (nivolumab and pembrolizumab) showed a pooled response rate of 17.3% (defined as halting of tumor progression, signs of tumor regression or tumor resolution) and overall survival of 10.4 months in patients with advanced HCC [92,93]. These rates are similar to those seen with sorafenib treatment but patients in this meta-analysis had less harsh side effect profiles [92,93]. Interestingly, one study noted differing responses whether HBV or HCV was the inciting disease, likely because HCV is an RNA virus and does not include any genomic integration [94]. While offering a more favorable side effect profile, early findings suggest that PD-1 inhibitors still offer only meager improvements in prognosis.

### 5.3. Cytotoxic T Lymphocyte-Associated Protein 4 (CTLA-4) Inhibitors

CTLA-4 or CD152, is an immune system checkpoint receptor which downregulates immune system response by limiting co-stimulation, activation and subsequent proliferation of T cells through competition against the normal co-stimulatory signal and were first studied in mesothelioma [95]. Tremelimumab is a CTLA-4 inhibitor studied in HCC patients who progressed on sorafenib and showed promising anti-tumor activity and an acceptable safety profile [96]. In a study of 20 patients given tremelimumab, the partial response rate or regression of tumor defined by the Response Evaluation Criteria in Solid Tumors (RECIST) criteria, was 18% (3/17). The disease control rate, defined as the percentage of patients who had stable disease, partial regression or complete resolution after an intervention was applied, was 76% (13/17) [96]. In addition, tremelimumab was studied in combination with liver-directed therapy (TACE and TARE) in BCLC stage B & C patients with a partial regression rate of 26% (5/19) but a 12 month tumor progression-free survival of 33.1% [97]. While highlighting the benefit of multi-modality treatment, CTLA-4 inhibitors require more extensive studies in a larger subset of patients to truly classify the HCC response as a success. Even then, little can be done to combat HCC tumors which are resistant or become resistant to this therapy.

## 6. Chimeric Antigen Receptor (CAR) T Cell Based Therapy

First identified as a means to study T cell activation, chimeric antigen receptors (CARs) fused to T cells were used by a group of scientists, including Carl June from the University of Pennsylvania, to successfully treat a 5 year old girl named Emily Whitehead with relapsing acute lymphoblastic leukemia (ALL) in 2010 [98]. CAR T cell therapy research increased following their success and was first approved by the FDA in 2017 for the treatment of ALL [98]. It has since been approved for treatment in certain types of non-Hodgkin lymphoma including aggressive, relapsed or refractory, diffuse large B cell lymphoma, primary mediastinal B-cell lymphoma, high grade B-cell lymphoma, transformed follicular lymphoma and mantle cell lymphoma [99]. This therapy involves isolating T cells from patients or donors and transfecting them with viral vectors containing specific chimeric antigen receptor sequences that recognize previously characterized tumor cell antigens [100,101]. The structure of the CAR consists of an extracellular antigen recognition domain derived from the variable portion of an antibody, a transmembrane domain and an intracellular domain that leads to signal transmission [102]. The extracellular domain provides the specificity as it can be designed against a specific antigen, while the potency of the immune response can be dictated by aggregation of the intracellular domains in the presence of high antigen concentrations. Once these CAR T cells are generated, they are returned back to the patient as an infusion. A graphic scheme of the process of harvesting, creating and treating patients with CAR T cells is shown in Figure 3. In addition, later iterations of CAR T cells have been developed to include their own co-stimulation signal, increasing their potency and ability to kill [102]. The promise of CAR T therapy lies in the ability to modulate therapy to target various, specific and predetermined antigens. Indeed, CAR T therapy for HCC has multiple proposed antigen targets, some of which are displayed in Figure 4. Their specifics as well as limitations are illustrated in the section below.

### 6.1. Targets of CAR T Therapy

#### 6.1.1. AFP/MHC complex

AFP is secreted and sometimes utilized as a tumor marker for HCC. Previously believed to be too non-specific and non-functional to target, all proteins are presented in class I MHCs and, thus, the opportunity to program CAR T cells to recognize this complex was established [103,104]. The issue with AFP is that expression levels do not necessarily correlate with stage of the disease or may not be present at all [105]. Furthermore, little is known or agreed upon about its actual mechanism or regulation, meaning that while this complex could be targeted, it could essentially be a decoy or byproduct that has no mechanistic or functional consequence or worse, cause off-target unintended detrimental effects [106].

#### 6.1.2. MUC1

Mucin-1 (MUC-1) is also known as polymorphic epithelial mucin and is a group of glycoproteins with high molecular mass that is overexpressed in human cancer cells including HCC cells [107]. A Phase I clinical trial of MUC-1 CAR T cells is currently underway for patients who have MUC-1 expressing HCC tumors (NCT02587689) [108]. No results are available from this trial yet. The limitation with MUC-1 as a target is that antigens are heterogeneously expressed within individual tumors and thus, the programming of CAR Ts in its current state may only target certain HCC tumors [109].

#### 6.1.3. EpCAM

Epithelial cell adhesion molecule (EpCAM) is a transmembrane glycoprotein involved in cell signaling, migration, proliferation and differentiation. It has been identified as a marker for stem/progenitor cells of the liver and is disproportionately upregulated in HCC [110]. For this reason, EpCAM may be a potential therapeutic target [111]. Two medical centers are currently testing EpCAM CAR T cells in the clinical setting, looking at their safety and efficacy profiles and as a treatment for HCC (NCT 03013712 and NCT 02729493) [112]. No results are available yet. Limitations are not yet known but the heterogeneity of HCC and its presenting antigens may prove resistant to this CAR T target.

#### 6.1.4. GPC-3

Glypican-3 (GPC3) is a member of heparan sulfate proteoglycans and is located on the cell surface. GPC3 is known to be upregulated in HCC as well as other solid tumors [113]. This protein contributes to carcinogenesis by acting as a co-receptor for a common oncologic cell proliferation pathway [114,115,116]. It has been studied as a tumor associated antigen as it is found in 70–80% of HCC cells but is not found on normal adult human cells, a feature that separates its utility from other markers of HCC [117]. Recently, a study showed that GPC3-targeted CAR T cells could eliminate orthotopic and xenograft hepatocellular carcinomas in vitro and in vivo [117,118,119]. GPC3-targeted CAR T cells could prove a promising therapy but additional verification for safety is currently underway with no results available yet (NCT02395250). Newer studies are looking to combine these CAR T cell infusions with other modes of delivery, such as intra-tumoral injection, to garner a greater sustained immune response but no results are available for this study yet either (NCT03130712).

### 6.2. Limitations of CAR T Therapy

In the treatment of advanced HCC cancer, CAR T seems to overcome many barriers that current therapies cannot. However, CAR T cell therapy creates a new set of obstacles that must be addressed. Concerns with this mode of therapy include a reported “on-target, off-tumor” destruction of normal cells as well as an exaggerated cytokine response when CAR T cells are infused, both of which can be extremely harmful to the patient [101,120]. In regards to healthcare cost, while autologous treatments do exist, efficient and economic production of CAR T therapy lies in an ‘off-the-shelf’ model in which healthy donor T cells are used. Consequently, the donor’s T cells must be characterized prior to T cell isolation and CAR T cell creation as not all recipients may be compatible, creating the opportunity for more patient harm [121]. Current work is being done to further refine the antigen specificity through structural changes as well as programming an inducible suicide switch in each CAR T cell to promote regulation to prevent some of these issues [122,123].

Additionally, while targets may be accessed via infusions in hematologic malignancies or cancers of circulating blood cells such ALL, solid tumors are static and surrounded by a tumor microenvironment. As we have discussed, HCC tumor cells are particularly adept at altering and evading immune responses, ultimately hindering CAR T immune cell trafficking and diminishing efficacy significantly [100,120]. Another issue involves the lack of consistent antigen presentation in solid tumors leading to ineffective target definition even within one etiology [101].

There has been parallel work done in the field of immunotherapy on the use of autogenously-generated anti-tumor lymphocytes in the treatment of advanced HCC. Preliminary studies using various HCC targets have been performed in patients, with moderately improved responses. Some studies reported increases in survival from 30 months to 44 months on average as well as increases in disease-free survival over 1-year (RR = 1.23), 2-year (RR = 1.37) and 3-years (RR = 1.35) in patients who received adjuvant autogenously-generated activated cytokine-induced killer T cells and natural killer cells after either surgical resection, radiofrequency ablation or percutaneous ethanol injection [124]. In addition, modest improvements in overall survival at 1-year (RR = 1.08), 2-years (RR = 1.14) and 3-years (RR = 1.15) are reported in a meta-analysis of that same patient population [125]. While showing the feasibility of immunotherapy for advanced HCC, these findings spotlight the need for additional refinement in CAR T therapy as the next generation of treatment. The promise of this modality is there is the potential for patient-specific programmable targeting, which can address the continued problem of HCC heterogeneity.

## 7. Multimodal Therapy

Single-modality therapy options each have their own set of limitations leading to decreased efficacy. Only 10–20% of treated HCC patients show a durable response to current treatment. However, a multimodal approach has been explored in HCC treatment, with the most promising combinations involving immunotherapy.

### 7.1. Systemic Therapies and Checkpoint Inhibitors

One strategy trialed was the combination of checkpoint inhibitors and systemic therapies, in the hopes of augmenting the effect of systemic therapy. PD-1 inhibitors, as well as other checkpoint inhibitors, have been trialed in combination with sorafenib and have shown increased anti-tumor effects in in vivo models. Unfortunately, in humans, disease-free and overall survival were not statistically increased and the combination requires more broad randomized control trials [126,127].

### 7.2. Locoregional Therapy

To overcome the aforementioned immune cell trafficking barriers of systemically infused CAR T therapies, techniques developed for TARE are being used to deliver CAR T cell infusions [100]. Highly selective catheterization of branches of the hepatic artery as well as injection directly into the tumor itself ensure optimal localization [128]. This could potentially eliminate the immune cell trafficking issue limiting CAR T therapy, as well as overcome the resistant HCC tumor microenvironment.

In addition, radiofrequency ablation and irradiation can be combined with either systemic or directly infused CAR T cell to improve the efficacy of CAR T cells by generating a localized immune response which leads to increased antigen presentation [129,130]. This approach has shown improvement in the native antitumor response compared to single-modality radiation therapy and has even been shown to “downstage” patients rendering them eligible for resection or transplant [131,132].

### 7.3. Post-Transplant

Though most immunotherapy options are presented to patients that are not surgical candidates, for the growing population of post-transplant patients there is a risk of developing recurrent HCC which would require systemic treatments. Immunosuppression, a requirement after transplant, has historically excluded the transplant population from clinical trials using immunotherapy due to concerns for decreased efficacy and increased harm in this patient population using this modality. PD-1 inhibitors were contraindicated in patients with previous solid organ transplantation as there is a documented risk of organ rejection, quoted to be 37.5% in liver transplants, in a systematic review [133]. However, more recently, there have been case reports suggesting a benefit in the use of PD-1 inhibitors in this population [134,135]. Currently, there is one clinical trial assessing the safety and efficacy of PD-1 inhibitors in patients with previous liver transplant that have recurrence of the disease which is not amenable to other therapeutic options (NCT 03966209). As this population continues to grow over the coming decades, information must be accumulated in which therapeutics are safe and efficacious for patients unfortunate enough to be afflicted by recurrent disease.

## 8. Conclusions

Hepatocellular carcinoma is a significant cause of morbidity and mortality worldwide, with no reliable method to detect the disease early in its progression. This is combined with the fact that systemic treatments available offer modest, at best, improvements in overall survival and disease-free survival in advanced or recurrent HCC, highlighting the need for novel therapeutic strategies to treat advanced stage HCC patients. To date, the most promising modality to offer this solution is immunotherapy.

While multi-modal treatment may offer augmented HCC response rates when tested properly, this option likely will not address the heart of the issue—HCC is a heterogeneous disease of multiple and often-times interwoven, etiologies. Its complex immunomodulation and tumor microenvironment require even more specific and individual treatments, something most obtainable by immunotherapy. The future of HCC treatment must strive for a patient-specific, tumor-specific treatment and do so in a reproducible and cost-effective manner.

If we can correctly identify and exploit druggable targets utilizing immunotherapy on an individual level, we can begin to slow disease progression, downstage patients to be eligible for surgical resection, combine it with surgical resection for improved results or eliminate the need for surgical resection altogether. In this regard, CAR T cell therapy, with its unlimited programmable potential, could be the key to unlock the personalized HCC treatment needed to treat this heterogeneous disease.

## Figures and Tables

**Figure 1 ijms-21-06757-f001:**
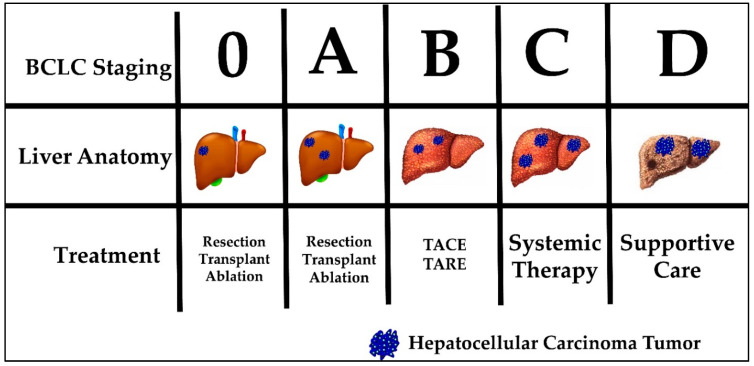
Barcelona Clinic Liver Cancer (BCLC) staging, liver anatomy, tumor burden and treatment.

**Figure 2 ijms-21-06757-f002:**
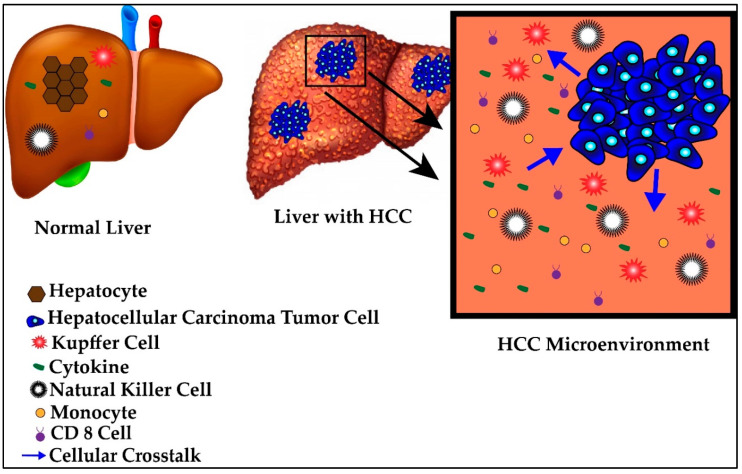
Normal versus hepatocellular carcinoma (HCC) microenvironment.

**Figure 3 ijms-21-06757-f003:**
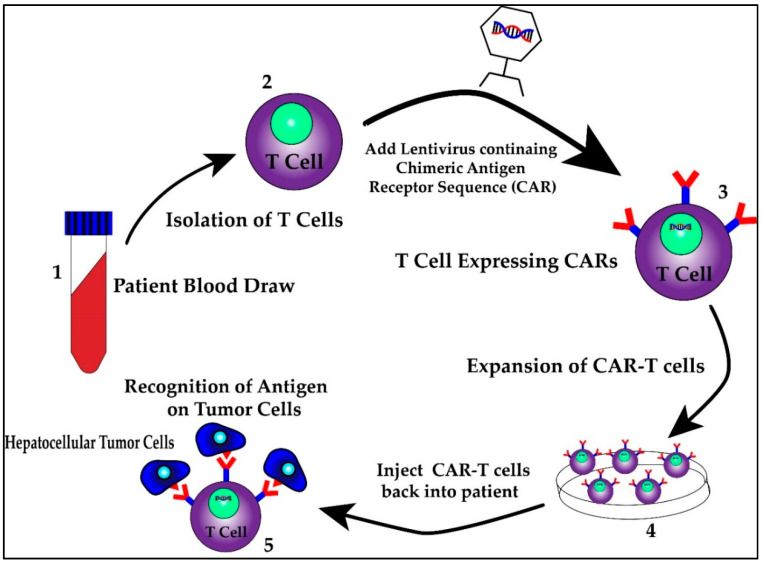
The process of harvesting, creating and treating patients with CAR T cells

**Figure 4 ijms-21-06757-f004:**
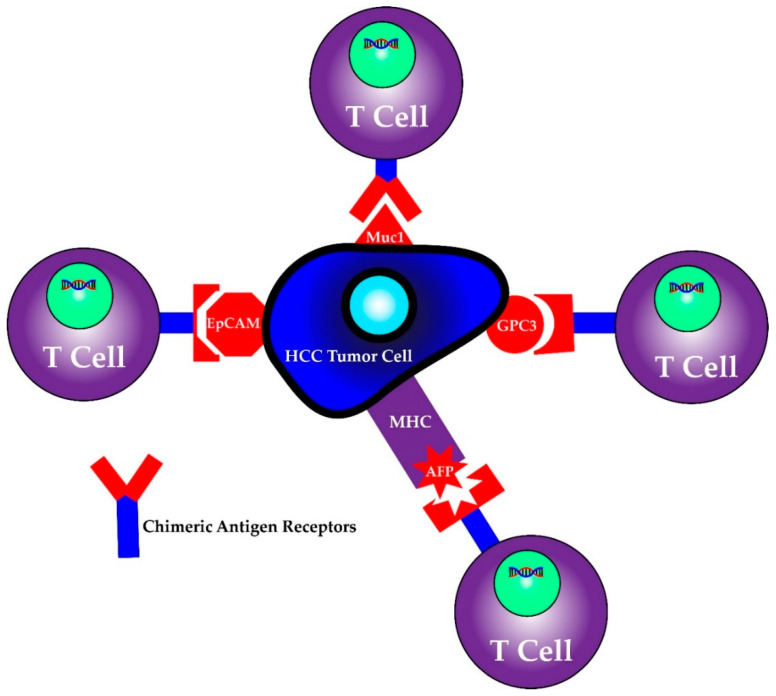
CAR T cells can be programmed to recognize individual antigens.

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
