# Peer review of "Hepatocellular Carcinoma—The Influence of Immunoanatomy and the Role of Immunotherapy"

_ijms, 2020, doi:10.3390/ijms21186757_

Round 1
Reviewer 1 Report
This is a review article regarding immunotherapy such as CAR T cell therapy for HCC, as well as current management of HCC. This manuscript is interesting and includes important and useful information for physicians involving HCC management.
- Since the authors mainly described CAR T cell therapy amongst several kinds of immunotherapy, the title of “Immunotherapy in Hepatocellular Carcinoma” is somewhat overstated. The authors should re-consider the title of this manuscript.
- The authors highlighted the unique immune-environment in the liver for the creation of the HCC microenvironment. However, differences between the HCC microenvironment and other tumor microenvironments have not been clearly demonstrated. Moreover, because of this unique microenvironment in HCC, does immunotherapy such as CAR T cell therapy provide different effects on HCC than other tumors?
- The authors stated that sorafenib is currently the only FDA approved systemic treatment for advanced HCC. Is it true?
Author Response
We thank you for your time in reviewing and commenting on the paper. Below are our responses to comments made with their associated corrections.
Point 1: Since the authors mainly described CAR T cell therapy amongst several kinds of immunotherapy, the title of “Immunotherapy in Hepatocellular Carcinoma” is somewhat overstated. The authors should re-consider the title of this manuscript.
Response 1: We have revised the title to "Hepatocellular Carcinoma: the Influence of Immunoanatomy and the Role of Immunotherapy."
Point 2: The authors highlighted the unique immune-environment in the liver for the creation of the HCC microenvironment. However, differences between the HCC microenvironment and other tumor microenvironments have not been clearly demonstrated. Moreover, because of this unique microenvironment in HCC, does immunotherapy such as CAR T cell therapy provide different effects on HCC than other tumors?
Response 2: While this is a credible and interesting question, no current data supports anything fundamentally different in the HCC tumor microenvironment versus other solid tumors. We have added a sentence at the end of section 4.2 "Tumor Microenvironment" to reflect this. It reads "While the liver itself does contain some unique features, more research is needed to define how this compares to liver microenvironments of other solid tumors or how it may influence the success or failure of immunotherapy options."
Point 3: The authors stated that sorafenib is currently the only FDA approved systemic treatment for advanced HCC. Is it true?
Response 3: There are other systemic therapies which have been approved, just not as well studied or considered in standard of care practice. Language throughout the manuscript has been updated to reflect this. In addition, in section 3.4 "Systemic Therapy" a sentence which reads "Other systemic medications have more recently been approved for HCC, however further studies will be required to elucidate their efficacy," has been added for clarification.
Reviewer 2 Report
Authors present a quality and well-written manuscript review that reviews immunotherapy in hepatocellular carcinoma. Authors summarize HCC, its staging system, current therapy, and immunotherapy medications currently being utilized or studied in the treatment of HCC. They also discuss current diagnosis and treatment of HCC, the unique contribution of the immunoanatomy of the liver in both its natural healthy and HCC disease state, and provide an overview of immunotherapy in HCC from the past and present with the hope of shedding light on where the field needs to expand in the future.
Comments:
- Line 356. Instead of “removing” better use “isolating”
- Line 375. Lentiviruses are retroviridae carrying RNA, not DNA. So better use term “chimeric antigen receptor sequence”
- Line 363. “given back” –> “returned back”
- Line 364. “graphic schematic” –> “graphic scheme”
- Figure 3 is incorrect. At step 2->3 lentivirus is added (not DNA). Step 3–>4 can be omitted, there is no need to show this. Step 4->5 not "replication", but "expansion".
- Authors are highly encouraged to cite the following article that overviews the challenges of applying CAR-T cell therapy for treatment of solid tumors. https://doi.org/10.3390/cancers12010125
Overall, the manuscript is valuable for the immunotherapy scientific community and should be accepted for publication after minor edits are made and Figure 3 is corrected.
Author Response
We thank you for your time in reviewing and commenting on the paper. Below are our responses to comments made with their associated corrections.
Point 1: Line 356. Instead of “removing” better use “isolating”
Response 1: This word has been substituted.
Point 2: Line 375. Lentiviruses are retroviridae carrying RNA, not DNA. So better use term “chimeric antigen receptor sequence”
Response 2: This section has been rephrased.
Point 3: Line 363. “given back” –> “returned back”
Response 3: This word has been substituted.
Point 4: Line 364. “graphic schematic” –> “graphic scheme”
Response 4: This word has been substituted.
Point 5: Figure 3 is incorrect. At step 2->3 lentivirus is added (not DNA). Step 3–>4 can be omitted, there is no need to show this. Step 4->5 not "replication", but "expansion".
Response 5: This figure has been updated per your recommendations.
Point 6: Authors are highly encouraged to cite the following article that overviews the challenges of applying CAR-T cell therapy for treatment of solid tumors. https://doi.org/10.3390/cancers12010125
Response 6: This source is now referenced and cited twice in section 6.2 "Limitations of CAR-T therapy".
Round 2
Reviewer 1 Report
The authors revised their manuscript according to reviewer's comments.